# RGFN: Synthesizable Molecular Generation Using GFlowNets

Michał Koziarski [* 1 2]  Andrei Rekesh [* 3]  Dmytro Shevchuk [* 3]  Almer van der Sloot [1 2]  Piotr Gaiński [4]
Yoshua Bengio [1 2]  Cheng-Hao Liu [1 5]  Mike Tyers [6 3]  Robert A. Batey [3 7]

## Abstract

In this paper, we propose an extension of the GFlowNet framework that operates directly in the space of chemical reactions, offering out-of-the-box synthesizability, while maintaining comparable quality of generated candidates. We demonstrate that with the proposed set of reactions and fragments, it is possible to obtain a search space of molecules orders of magnitude larger than existing screening libraries while offering low costs of synthesis. We also show that the approach scales to very large fragment libraries, further increasing the number of potential molecules. Our experiments showcase the effectiveness of the proposed approach across a range of oracle models.

## 1. Introduction

In this paper, we propose Reaction-GFlowNet (RGFN), an extension of the GFlowNet framework (Bengio et al., 2023) that generates molecules by combining basic chemical fragments using a chain of reactions. We propose a relatively small collection of cheap and accessible chemical fragments using a chain of reactions, based upon established high-yield chemical transformations, that together still produce a search space orders of magnitude larger than existing chemical libraries. We additionally propose several domain-specific extensions of the GFlowNet framework for state representation and scaling to a larger space of possible actions. We experimentally evaluate RGFN on a set of diverse screening tasks, including docking score approximation with a trained proxy model for soluble epoxide hydrolase (sEH), GPU-accelerated direct docking score calculations for multiple protein targets (Mpro, ClpP, TBLR1

and sEH), and biological activity estimation with a trained proxy model for senolytic classification (Wong et al., 2023). We demonstrate that RGFN produces similar optimization quality and diversity to existing fragment-based approaches while ensuring synthesizability out-of-the-box. Our analysis further indicates that the generated molecules exhibit a diverse range of chemical properties, as well as target specificity, demonstrating sufficient expressivity of the proposed fragments and synthesis pathways.

## 2. Related work

**Generative models for molecular discovery.** There exists a plethora of methods for molecular generation (Meyers et al., 2021; Bilodeau et al., 2022) using machine learning. They can be categorized depending on the molecular representation used: textual representation such as SMILES (Kang & Cho, 2018; Arús-Pous et al., 2020; Kotsias et al., 2020), molecular graphs (Jin et al., 2018; Maziarka et al., 2020; Pedawi et al., 2022) or 3D atom coordinate representations (O Pinheiro et al., 2024); as well as the underlying methodology, e.g., variational autoencoders (Jin et al., 2018; Maziarka et al., 2020), reinforcement learning (Pedawi et al., 2022; Korablyov et al., 2024) or diffusion models (Runcie & Mey, 2023). Recently, Generative Flow Networks (GFlowNets) (Bengio et al., 2021; Nica et al., 2022; Roy et al., 2023; Shen et al., 2023; Volokhova et al., 2024) have emerged as a promising paradigm for molecular generation due to their ability to sample diverse candidate molecules, which is crucial in the drug discovery process. Traditionally, GFlowNets operated on the graph representation level, and candidate molecules were generated as a sequence of actions in which either individual atoms or small molecular fragments were combined to form a final molecule. While using graph representations, as opposed to textual or 3D representations, allows the enforcement of the validity of the generated molecules, it doesn't guarantee a valid route by which to synthesize them. This work expands on the GFlowNet framework by modifying the space of actions to consist of choosing molecular fragments and executing compatible chemical reactions/transformations, in turn guaranteeing both validity and synthesizability.

**Synthesizability in generative models.** One approach to

---

[*]Equal contribution [1]Mila – Québec AI Institute [2]Université de Montréal [3]University of Toronto [4]Jagiellonian University [5]McGill University [6]The Hospital for Sick Children Research Institute [7]Acceleration Consortium. Correspondence to: Michał Koziarski <michal.koziarski@mila.quebec>, Andrei Rekesh <a.rekesh@mail.utoronto.ca>, Dmytro Shevchuk <dmytro.shevchuk@mail.utoronto.ca>.

*Accepted at the 1st Machine Learning for Life and Material Sciences Workshop at ICML 2024.* Copyright 2024 by the author(s).

ensuring the synthesizability of generated molecules is by using a scoring function, either utilizing it as one of the optimization criteria (Korablyov et al., 2024), or as a postprocessing step by which to filter generated molecules. Multiple scoring approaches, both heuristic (Ertl & Schuffenhauer, 2009; Genheden et al., 2020) and ML-based (Liu et al., 2022), exist in the literature. However, synthesizability estimation is difficult in practice. It can fail to generalize out of distribution in the case of ML models, may significantly reduce the number of high-scoring candidates, and does not necessarily account for the cost of synthesis. Because of this, a preferable approach might be to constrain the space of possible molecules to those easily synthesized by operating in a predefined space of chemical reactions and fragments. Several existing works employ this approach (Bradshaw et al., 2019; Gao et al., 2021; Swanson et al., 2024), including reinforcement learning-based methods (Gottipati et al., 2020; Horwood & Noutahi, 2020), which are conceptually closest to this paper. We extend this line of work not only by translating it to the GFlowNet framework but also by proposing a curated set of robust chemical reactions and fragments that ensure efficient synthesis at lower total costs.

## 3. Method

### 3.1. Generative Flow Networks

GFlowNets are amortized variational inference algorithms that are trained to sample from an unnormalized target distribution over compositional objects. GFlowNets aim to sample objects from a set of terminal states $\mathcal{X}$ proportionally to a reward function $\mathcal{R} : X \to \mathbb{R}^+$. GFlowNets are defined on a pointed directed acyclic graph (*DAG*), $G = (S, A)$, where:

- $s \in S$ are the nodes, referred to as states in our setting, with the special starting state $s_0$ being the only state with no incoming edges, and the terminal states $\mathcal{X}$ have no outgoing edges,

- $a = s \to s' \in A$ are the edges, referred to as actions in our setting, and correspond to applying an action while in a state $s$ and landing in state $s'$.

A state sequence $\tau = (s_0 \to s_1 \to \ldots \to s_n = x)$, with $s_n = x \in \mathcal{X}$ and $a_i = (s_i \to s_{i+1}) \in A$ for all $i$, is called a complete trajectory. We denote the set of trajectories as $\mathcal{T}$.

### 3.2. Reaction-GFlowNet

Reaction-GFlowNet generates molecules by combining basic chemical fragments using a chain of reactions. The generation process is illustrated in Figure 1:

In the rest of this section, we describe the design of the

Reaction-GFlowNet in detail.

**Preliminaries**. Reaction-GFlowNet uses a predefined set of reaction patterns and molecules introduced in Section 3.3. We denote them as $R$ and $M$ respectively. As a backbone for our forward policy $P_F$, we use a graph transformer model $f$ from (Yun et al., 2019). The graph transformer takes as an input a molecular graph $m$ and outputs the embedding $f(m) \in \mathbb{R}^D$, where $D$ is the embedding dimension. In particular, $f$ can embed an empty graph . It can additionally be conditioned on the reaction $r \in R$ which we denote as $f(m, r)$. The reaction in this context is represented as its index in the reaction set $R$.

**Select an initial fragment**. At the beginning of each trajectory, Reaction-GFlowNet selects an initial fragment from $M$. The probability of choosing $i$-th fragment $m_i$ is equal to:

$$p(m_i|\emptyset) = \sigma^{|M|}(\mathbf{s})_i, \ s_i = \mathrm{MLP}_M(f(\emptyset))_i, \quad (1)$$

where $\mathrm{MLP}_M : \mathbb{R}^D \to \mathbb{R}^{|M|}$ is a multi-layer perceptron (MLP). The $\sigma^k$ is a standard softmax over the logits vector $\mathbf{s} \in \mathbb{R}^k$ of the length $k$:

$$\sigma^k(\mathbf{s})_i = \frac{\exp(s_i)}{\sum_{j=1}^{k} \exp(s_j)}.$$

**Select the reaction template**. The next step is to select a reaction that can be applied to the molecule $m$. The probability of choosing $i$-th reaction from $R$ is described as:

$$p(r_i|m) = \sigma^{|R|+1}(\mathbf{s})_i, \ s_i = \mathrm{MLP}_R(f(m))_i, \quad (2)$$

where $\mathrm{MLP}_R : \mathbb{R}^D \to \mathbb{R}^{|R|+1}$ is an MLP that outputs logits for reactions from $R$ and an additional stop action with index $|R| + 1$. Choosing the stop action in this phase ends the generation process. Note that not all the reactions may be applied to the molecule $m$. We appropriately filter those reactions and assume that the score $s_i$ for them is equal to $-\infty$.

**Select another reactant**. We want to find a molecule $m_i \in M$ that will react with $m$ in the reaction $r$. The probability for selecting $m_i$ is defined as:

$$p(m_i|m, r) = \sigma^{|M|}(\mathbf{s})_i, \ s_i = \mathrm{MLP}_M(f(m, r))_i \quad (3)$$

where $\mathrm{MLP}_M$ is shared with the initial fragment selection phase. As in the previous phase, not all the fragments can be used with the reaction $r$, so we filter them out.

**Perform the reaction and select one of the resulting molecules**. In this step, we apply the reaction $r$ to the two fragment molecules chosen in previous steps. As the reaction pattern can be matched to multiple parts of the molecules, the result of this operation is a set of possible

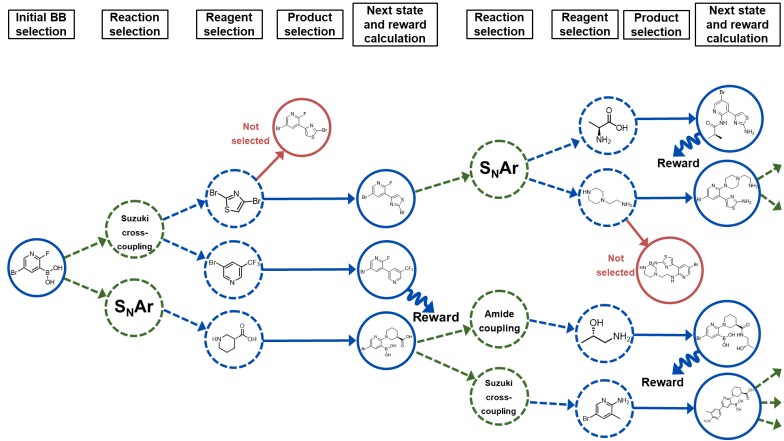

*Figure 1.* Illustration of RGFN sampling process. At the beginning, the RGFN selects an initial molecular building block. In the next two steps, a reaction and a proper reagent are chosen. Then the reaction is performed and one of the resulting molecules is selected. The process is repeated until the stop action is chosen. The obtained molecule is then evaluated using the reward function.

outcomes $M'$. We choose the molecule $m'i \in M'$ by sampling from the following distribution:

$$p(m'_i|r) = \sigma^{|M'|}(\mathbf{s})_i, \; s_i = \mathrm{MLP}_{M'}(f(m'_i, r)), \quad (4)$$

where $\mathrm{MLP}_{M'} : \mathbb{R}^D \to \mathbb{R}$ scores the embedded $m'_i$ molecule.

**Backward Policy.** A backward policy in RGFN is only non-deterministic in states corresponding to a molecule $m$ which is a result of performing some reaction $r \in R$ on molecule $m'$ and reagent $m'' \in R$. We denote the set of such tuples $(r, m', m'')$ that may result in $m$ as $T$. We override the indexing and let $(r_i, m'_i, m''_i)$ be the $i$-th tuple from $T$. The probability of choosing the $i$-th tuple is:

$$p((r_i, m'_i, m''_i)|m) = \sigma^{|T|}(\mathbf{s})_i, \; s_i = \mathrm{MLP}_B(f(m'_i, r_i)), \quad (5)$$

where $\mathrm{MLP}_B : \mathbb{R}^D \to \mathbb{R}$ and $f$ is a backbone transformer model similar to the one used in the forward policy. To properly define $T$, we need to implicitly keep track of the number of reactions performed to obtain $m$ (denoted as $k$). Only those tuples $(r, m', m'')$ are contained in the $T$ for which we can recursively obtain $m'$ in $k - 1$ reactions.

**Action Embedding.** While the $\mathrm{MLP}_M$ used to predict the probabilities of selecting a molecule $m_i \in M$ works well for our predefined $M$, it underperforms when the size of possible chemical fragments is increased. Our intuition is that such an $\mathrm{MLP}_M$ struggles to reconstruct the relationship between the molecules. Intuitively, when a molecule $m_i$ is chosen in some trajectory, the training signal from the loss function should also influence the probability of choosing a structurally similar $m_j$. However, the $\mathrm{MLP}_M$ disregards the structural similarity by construction and it intertwines the probabilities of choosing $m_i$ and $m_j$ only with the softmax function. To incorporate the relationship between molecules

into the model, we embed the molecular fragments with a simple machine learning model $g$ and reformulate the probability of choosing a particular fragment $m_i$:

$$p(m_i|m, r) = \sigma^{|M|}(\mathbf{s})_i, \; s_i = \phi(Wf(m, r))^T g(m_i), \quad (6)$$

where $\phi$ is some activation function (we use GELU) and $W \in \mathbb{R}^{D \times D}$ is a learnable linear layer. Note that if we define $g(m_i)$ as an index embedding function that simply returns a distinct embedding for every $m_i$, we will obtain a formulation equivalent to Equation (3). To leverage the structure of molecules during the training, we use $g$ that linearly embeds a (MACSS) fingerprint of an input molecule $m_i$ along with the index $i$. Note that this approach does not add any additional computational costs during the inference as the embeddings $g(m_i)$ can be cached. In Appendix C.2, we show that this method greatly improves the performance when scaling to larger sets of fragments.

### 3.3. Chemical language

Seventeen reactions and 350 building blocks were selected for our first-generation model. The reactions used include amide bond formation, nucleophilic aromatic substitution, Michael addition, isocyanate-based urea synthesis, sulfur fluoride exchange (SuFEx), sulfonyl chloride substitution, alkyne-azide and nitrile-azide cycloadditions, esterification reactions, urea synthesis using carbonyl surrogates, Suzuki-Miyaura, Buchwald-Hartwig, and Sonogashira cross-couplings, amide reduction, and peptide terminal thiourea cyclization reactions to produce iminohydantoins and tetrazoles. The chosen reactions are known to be typically quite robust and generally high-yielding (75-100%), thus enforcing reliable synthesis pathways when sampling molecules from our model. During the construction of the curated building block database, only affordable

reagents (building blocks) were considered. For the purposes of this study we define affordable reagents to be those priced at less than or equal to $200 per gram. The mean cost per gram of reagents is $22.52, the lowest cost $0.023 per gram, and the highest cost $190 per gram.

A crucial consideration when choosing the set of reactions and fragments used is the state space size. This is difficult to compute precisely since a different set of reactions or fragments is valid for every state in trajectory. We estimate this based on 1,000 random trajectories instead (details can be found in Appendix A). In addition to our 350 low-cost fragments, we also perform this analysis with 8,000 additional random Enamine fragments. Comparison for different numbers of maximum reactions is presented in Figure 2. As can be seen, even with curated low-cost fragments and limiting the number to a maximum of four reactions, state space size is an order of magnitude greater than Enamine REAL (Enamine, 2024). This size can increase significantly with the addition of more fragments and/or an increase in the maximum number of reactions. Additional discussion of scaling can be found in Appendix C.2.

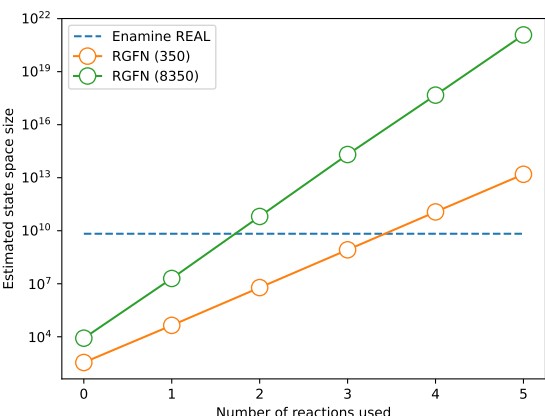

*Figure 2.* Estimation of the state space size of RGFN as a function of the maximum number of allowed reactions. RGFN (350) indicates a variant using 350 hand-picked inexpensive building blocks, while RGFN (8350) also uses 8,000 randomly selected Enamine building blocks. Enamine REAL (6.5B compounds) is shown as a reference.

## 4. Experimental study

In the conducted experiments we compare oracle scores and synthesizability scores of RGFN with several state-of-the-art reference methods. Then, we perform an in-depth examination of produced ligands across several biologically relevant targets. Experimental details can be found in Appendix B. Additional experiments examining the number of modes discovered by different methods and the capabilities of RGFN to scale to larger fragment libraries can be found

in Appendix C.

### 4.1. Comparison with existing methods

We begin experimental evaluation with a comparison to several state-of-the-art methods for molecular discovery. Specifically, we consider a genetic algorithm operating on molecular graphs (GraphGA) (Jensen, 2019) as implemented in (Brown et al., 2019), which has been demonstrated to be a very strong baseline for molecular discovery (Gao & Coley, 2020), Monte Carlo tree search-based SyntheMol (Swanson et al., 2024), and a fragment-based GFlowNet (FGFN) (Bengio et al., 2021) as implemented in (Recursion, 2024). Training details can be found in Appendix B.6. It is worth noting that besides SyntheMol, which also operates in the space of chemical reactions and building blocks derived from the Enamine database, our remaining benchmarks do not explicitly enforce synthesizability when generating molecules. Because of this, in this section, we will examine not only the quality of generated molecules in terms of optimized properties but also their synthesizability. We consider only a single reaction-based approach, as other existing methods employing this paradigm (Horwood & Noutahi, 2020; Gottipati et al., 2020) do not share code or curated reactions and fragments, making reproduction difficult.

We first examine the distributions of rewards found by each method across three different oracles used for training: sEH proxy, senolytic proxy, and GPU-accelerated docking for ClpP. The results are presented in Figure 3. As can be seen, while RGFN underperforms in terms of average reward when compared to the method not enforcing synthesizability (GraphGA), it outperforms SyntheMol's reaction-based sampling. Interestingly, when compared to standard FGFN, RGFN either performs similarly (ClpP docking) or achieves higher average rewards. This is most striking in the case of the challenging senolytic discovery task, in which a proxy is trained on a severely imbalanced dataset with less than 100 actives, resulting in a sparse reward function. We suspect that this, possibly combined with a lack of compatibility between the FGFN fragments and known senolytics, led to the failure to discover any high-reward molecules. However, RGFN succeeds in the task and finds a wide range of senolytic candidates.

Finally, we focus on the synthesizability of generated compounds. We present average values of several synthesizability-related metrics, computed over top-k modes generated for each method, in Table 1. For completeness, we also include SAScores (Ertl & Schuffenhauer, 2009), but note that they are only a rough approximation of ease of synthesis. For a better estimate of synthesizability we perform retrosynthesis using AiZynthFinder (Genheden et al., 2020) and count the average number of molecules for which a

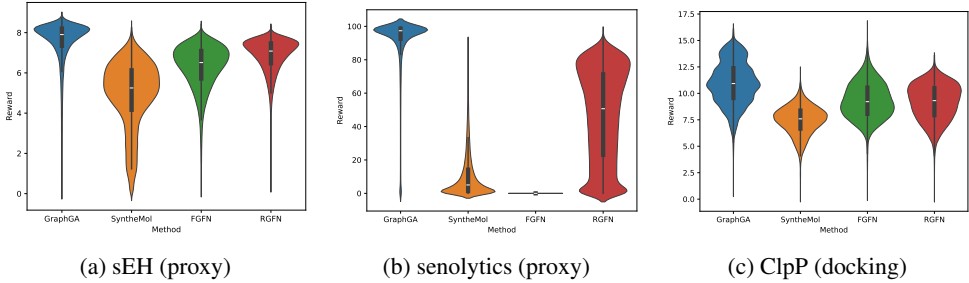

| (a) sEH (proxy) | (b) senolytics (proxy) | (c) ClpP (docking) |

*Figure 3.* Distributions of rewards across different tasks.

valid retrosynthesis pathway was found. Note that to reduce variance, we compute SAScores over top-500 modes, but due to high computational cost, AiZynth scores are computed only over top-100 modes. As can be seen, while there is some variance across tasks, RGFN performs similarly to SyntheMol in terms of both synthesizability scores, and significantly outperforms GraphGA and FGFN. All RGFN modes were additionally inspected manually by a chemist and confirmed as synthesizable, which indicates that AiZynth scores are likely underestimated.

*Table 1.* Average values of synthesizability-related metrics.

| Task | Method | SAScore ↓ | AiZynth ↑ |
|------|--------|-----------|-----------|
| sEH | GraphGA | 3.87 ± 0.24 | 0.04 |
| | SyntheMol | 2.85 ± 0.55 | 0.80 |
| | FGFN | 3.43 ± 0.48 | 0.14 |
| | RGFN | 3.09 ± 0.39 | 0.56 |
| Seno. | GraphGA | 2.92 ± 0.26 | 0.05 |
| | SyntheMol | 2.77 ± 0.40 | 0.53 |
| | FGFN | 3.74 ± 0.54 | 0.01 |
| | RGFN | 3.24 ± 0.32 | 0.58 |
| ClpP | GraphGA | 4.14 ± 0.51 | 0.00 |
| | SyntheMol | 2.86 ± 0.56 | 0.56 |
| | FGFN | 2.94 ± 0.54 | 0.25 |
| | RGFN | 2.83 ± 0.22 | 0.65 |

### 4.2. Examination of the produced ligands

In the final stage of experiments we examine the capabilities of RGFN to produce high quality binders to a set of diverse protein targets. The aim is to evaluate whether 1) the chemical language used is expressive enough to produce structurally diverse molecules for different targets, and 2) whether generated ligands form realistic poses in the binding pockets. We first demonstrate the diversity of ligands across targets on a UMAP plot of extended-connectivity fingerprints Figure 4. Ligands assigned to specific targets form very distinct clusters, showcasing their diversity. Interestingly, we observe structural differences between sEH proxy and sEH docking, possibly indicating poor approximation of docking scores by the proxy model. Secondly, we examine

the docking poses of the highest scoring generated ligands. As can be seen, the generated molecules produce realistic docking poses, closely resembling those of known ligands (Appendix D). Overall, this demonstrates the usefulness of the proposed approach in the docking-based screens.

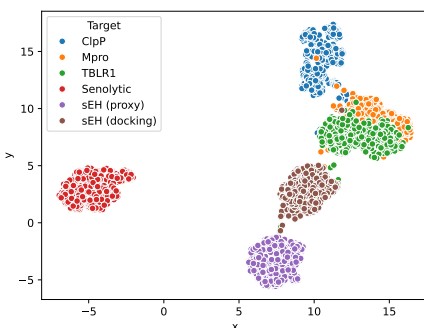

*Figure 4.* UMAP plot of chemical structures of top-500 modes generated for each target. RGFN generates sufficient chemical diversity to produce distinct clusters of compounds.

## 5. Conclusions

In this paper, we present RGFN, an extension of the GFlowNet framework that operates in the action space of chemical reactions. We propose a curated set of high-yield chemical reactions and low-cost molecular fragments that can be used with the method. We demonstrate that even with this small set of reactions and fragments, the proposed approach produces a state space with a size orders of magnitude larger than typical screening libraries while providing high synthesizability of generated compounds. We also show that the size of the search space can be further increased by including additional fragments and that the proposed action embedding mechanism improves scalability to very large fragment spaces.

We show that RGFN achieves roughly comparable average rewards to state-of-the-art methods, and it outperforms another approach operating directly in the space of chemical re-

actions and, crucially, standard fragment-based GFlowNets. At the same time, it significantly improves the synthesizability of generated compounds when compared to a fragment-based GFlowNet. Conducted analysis of ligands produced across the set of diverse tasks demonstrates sufficient diversity of proposed chemical space to generalize to various targets. While difficult to demonstrate experimentally, ease of synthesis (due to the small stock of cheap fragments and high-yield chemical reactions used) combined with reasonably high optimization quality offer a promising direction for high-throughput screening applications.

## Acknowledgements

This work was supported by funding from CQDM Fonds d'Accélération des Collaborations en Santé (FACS) / Acuité Québec and the National Research Council (NRC) Canada, the Canadian Institutes for Health Research (CIHR), Samsung and Microsoft. Computational resources were provided by the Digital Research Alliance of Canada (https://alliancecan.ca) and Mila (https://mila.quebec). The research of P. Gainski was supported by the National Science Centre (Poland), grant no. 2022/45/B/ST6/01117. We gratefully acknowledge Poland's high-performance Infrastructure PLGrid (ACK Cyfronet Athena, HPC) for providing computer facilities and support within computational grant no PLG/2023/016550. Additional thanks for funding provided to the University of Toronto's Acceleration Consortium from the Canada First Research Excellence Fund (Grant number - CFREF-2022-00042).

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

# A. State space size estimation

We estimate the state space size by first sampling 1,000 random trajectories, masking out the end-of-sequence action unless the maximum trajectory length $max$ is reached. Then, for every $i$-th reaction or fragment in the trajectory, we count the average number of valid fragments $frag_i$ and reactions $react_i$ from a given state in the trajectory, as well as the average number of unique trajectories $traj_i$ into which a state can be decomposed using backward policy. We estimate the state space size as

$$\frac{(\prod_{i=0}^{max} frag_i)(\prod_{i=1}^{max} react_i)}{traj_{max}}. \tag{7}$$

Experimentally derived average values of these parameters can be found in Table 2. Note that in the second setting we randomly picked 8,000 fragments from the Enamine stock (with the same balancing procedure as in Appendix C.2), which after merging with our own fragments, canonization and duplicate removal yielded a total of 8,317 fragments.

# B. Experimental details

## B.1. Set-up

Throughout the course of the conducted experimental study, we aim to evaluate the performance of the proposed approach across several diverse biological oracles of interest. This includes proxy models (machine learning oracles, pre-trained on the existing data and used for higher computational efficiency): first, the commonly used sEH proxy as described in (Bengio et al., 2021). Second, a graph neural network trained on the biological activity classification task of senolytic (Wong et al., 2023) recognition. Details of proxy models are provided in Appendix B.2.

Per the GFlowNet training algorithm, the reward is calculated for a batch of dozens to hundreds of molecules at each training step, rendering traditional computational docking score algorithms like AutoDock Vina (Trott & Olson, 2010) infeasible for very large training runs. As a result, previous applications of GFlowNets to biological design (Bengio et al., 2021; Shen et al., 2023) employed a fast pre-trained proxy model trained on docking scores instead. These proxies, while lightweight, present potential issues should the GFlowNet generate molecules outside their training data distributions and require receptor-specific datasets. To circumvent this, we use the GPU-accelerated Vina-GPU 2.1 (Tang et al., 2023) implementation of the QuickVina 2 (Alhossary et al., 2015) docking algorithm to calculate docking scores directly in the training loop of RGFN. This approach allows for drastically increased flexibility in protein target selection while eliminating proxy generalization failure. We select human soluble epoxy hydrolase (sEH), ATP-dependent Clp protease proteolytic subunit (ClpP), SARS-CoV-2 main protease (Mpro), and transducin $\beta$-like-related protein 1 (5NAF) as targets for evaluating RGFN using a docking reward.

Training details for all of the generative methods can be found in Appendix B.6.

## B.2. Proxy models

The sEH proxy is described in (Bengio et al., 2021). It is an MPNN trained on a normalized docking score data. We utilize the exact same model checkpoint as provided in (Recursion, 2024).

Senolytic classification model is a graph neural network trained on the biological activity classification task of senolytic (Wong et al., 2023) recognition. Specifically, it was trained on two combined, publicly available senolytic datasets (Wong et al., 2023; Smer-Barreto et al., 2023). Reward is given by the predicted probability of a compound being a senolytic. It is worth noting that due to the low amount of data and high imbalance ($<$ 100 active compounds, a high proportion of which contained macrocycles

Table 2. Experimentally derived average values of valid fragments, valid reactions, and possible trajectories.

| # reactions | 350 fragments | | | 8350 fragments | | |
|---|---|---|---|---|---|---|
| | $frag_i$ | $react_i$ | $traj_i$ | $frag_i$ | $react_i$ | $traj_i$ |
| 0 | 350.0 | - | 1.0 | 8317.0 | - | 1.0 |
| 1 | 37.5 | 11.8 | 3.5 | 835.8 | 12.0 | 4.2 |
| 2 | 39.9 | 16.5 | 16.8 | 822.1 | 15.9 | 17.0 |
| 3 | 40.7 | 15.4 | 76.7 | 832.6 | 17.0 | 75.2 |
| 4 | 40.0 | 15.8 | 349.3 | 814.0 | 18.0 | 480.8 |
| 5 | 42.1 | 16.8 | 1825.8 | 857.6 | 18.9 | 3058.1 |

and were infeasible to construct with fragment-based generative models), this is expected to be a difficult task with sparse reward.

The senolytic proxy model consisted of 5 GIN layers (Xu et al., 2018a) with hidden dimensionality of 500, utilized Jumping Knowledge shortcuts (Xu et al., 2018b), and had a single output MLP layer. Pretraining was done in an unsupervised fashion on the ZINC15 dataset (Sterling & Irwin, 2015). The training was done for 30 epochs using Adam optimizer with a learning rate of $5 \times 10^{-5}$ and batch size of 50.

### B.3. GPU-accelerated docking

Our docking oracle first accepts canonized SMILES strings as input. These are then converted to RDKit Molecules, protonated, and a low-energy conformer is generated and minimized with the ETKDG (Riniker & Landrum, 2015) conformer generation method and UFF (Rappe et al., 1992) force field, respectively. For computational efficiency, we generate one initial conformer per ligand. Each conformer is converted to a pdbqt file and docked against a target with Vina-GPU 2.1 using model defaults: exhaustiveness (denoted by "thread" in the implementation) of 8000 and a heuristically determined search depth $d$ given by

$$d = \max\left(1, \lfloor 0.36 \times N_{atom} + 0.44 \times N_{rot} - 5.11 \rfloor\right),$$
$$(8)$$

where $N_{atom}$ and $N_{rot}$ are the number of atoms and the number of rotatable bonds, respectively, in the generated molecule. Box sizes were determined individually to encompass each target binding site and centroids were calculated to be the average position of ligand atoms in the receptor PDB. A negative score is calculated and returned as a reward.

### B.4. Target preprocessing

Each target was prepared by removing its complexed inhibitor and atoms of other solvent or solute molecules. We selectively prepared the ClpP 7UVU protein structure by retaining only two monomeric units to ensure the presence of a single active site available for ligand binding and similarly

prepared the Mpro 6W63 protein structure by retaining only one monomeric unit.

### B.5. Ligand postprocessing

To ensure the diversity, specificity, and conformer validity of top-generated molecules for each target, we initially categorized our molecules into distinct modes, each representing any SMILES string with a Tanimoto similarity of 0.5 or lower with all other modes. Subsequently, we selected the top 100 modes based on their Vina-GPU 2.1 scores and filtered their docked poses using PoseBusters (Buttenschoen et al., 2023) in "mol" mode, where any pose failing any PoseBusters check was excluded from consideration. As a final precaution, we selected only modes with Tanimoto coefficients to known aggregators of 0.4 or lower using UCSF's Aggregation Advisor (Irwin et al., 2015) dataset. This process resulted in 35, 68, 31, and 15 top modes for sEH, ClpP, Mpro, and TBLR1 binders, respectively (Appendix E). Comparative analyses of docked top RGFN modes and confirmed sEH, ClpP, and Mpro ligand poses can be found in Appendix D. TBLR1 was omitted from the analysis due to a lack of known small-molecule ligands.

### B.6. Model training

Both RGFN and FGFN were trained with trajectory balance loss (Malkin et al., 2022) using Adam optimizer with a learning rate of $1 \times 10^{-3}$, logZ learning rate of $1 \times 10^{-1}$, and batch size of 100. The training lasted 4,000 steps. A random action probability of 0.05 and a replay buffer of 20 samples per batch were used. Both methods use graph transformer policy with 5 layers, 4 heads, and 64 hidden dimensions. Exponentiated reward $R(x) = exp(\beta * score(x))$ was used, with $\beta$ dependent on the task: 8 for sEH proxy, 0.5 for senolytic proxy, and 4 for all docking runs. Note that due to different ranges of score values, this resulted in a roughly comparable range of reward values.

All sampling algorithms were outfitted with the Vina GPU-2.1 docking, senolytic proxy, and sEH proxy scoring func-

tions. While model architecture hyperparameters and batch sizes were kept consistent between FGFN and RGFN, we allowed FGFN a maximum fragment count of 6 as opposed to RGFN's 5 due to RGFN's larger average building block sizes.

GuacaMol's Graph GA model was trained with a population size of 100, offspring size of 200, and a mutation rate of 0.01 for 2000 generations for a total of 400,000 visited molecules.

For SyntheMol experiments, we used the default building block library of 132,479 compatible molecules and pre-computed docking, senolytic, and sEH proxy scores for all prior to executing rollouts to follow the established methodology. Due to CPU constraints, sampling 500,000 molecules with SyntheMol was impractical. Instead, we executed 100,000 rollouts over approximately 72 hours to match the RGFN training time with docking, yielding 111,964 unique molecules. Additionally, we performed 50,000 rollouts each (approximately 30 hours) for sEH and senolytic proxies, resulting in 73,941 and 69,652 unique molecules, respectively.

## C. Additional experiments

### C.1. Number of discovered modes

We examine the number of discovered modes for each method, with a mode defined as a molecule with computed reward above a threshold (sEH: 7, senolytics: 50, ClpP docking: 10), and Tanimoto similarity to every other mode $< 0.5$. The number of discovered modes across tasks as a function of normalized iterations is presented in Appendix C.1. Note that in the case of GraphGA, FGFN, and RGFN this simply translates to the number of oracle calls, but for SyntheMol, due to large computational overhead, we impose a maximum number of rollouts such that training time was comparable to RGFN (see Appendix B.6 for details). As can be seen, despite slightly worse average rewards, FGFN still outperforms other methods in terms of the number of discovered modes (with the exception of senolytic discovery task, where it fails to discover any high-reward molecules). This suggests that RGFN samples are less diverse, possibly due to the relatively small number of fragments and reactions used. However, RGFN still outperforms remaining methods across all tasks, suggesting that it preserves some of the benefits of the diversity-focused GFlowNet framework.

### C.2. Scaling to larger sets of fragments

Next we investigate the influence of a fragment embedding scheme proposed in Section 3.2. In the standard implementation of the GFlowNet policy, actions are represented as independent embeddings in the MLP. These encode actions as indices, effectively disregarding their respective internal structures and all information contained therein. If there

is a helpful structure within the actions, the independent embeddings will need to learn it from scratch. While this may be a relatively easy task for small action spaces, it becomes more difficult when the size of the action space increases. To scale RGFN to a larger size of the building block library, we proposed to encode fragment actions using molecular fingerprints, allowing the model to leverage the internal structure of the actions without any additional computational overhead during the inference. In Figure 6, we observe that our fingerprint embedding scheme allows for drastically faster convergence compared to the standard independent action embedding, especially for large library sizes.

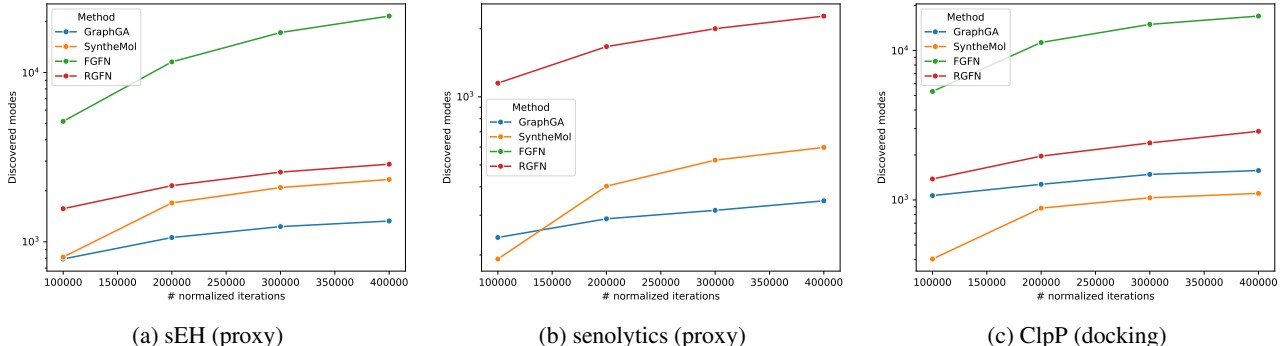

(a) sEH (proxy)

(b) senolytics (proxy)

(c) ClpP (docking)

*Figure 5.* Number of discovered modes as a function of normalized iterations. Log scale used.

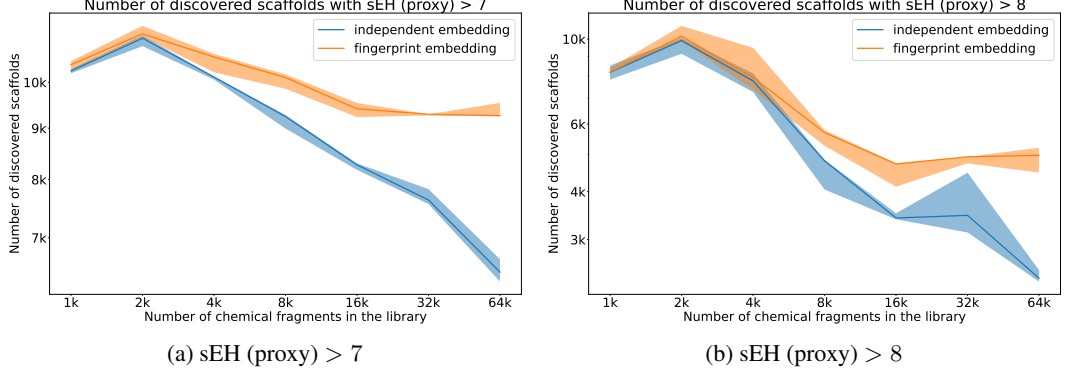

(a) sEH (proxy) > 7

(b) sEH (proxy) > 8

*Figure 6.* The number of discovered scaffolds with sEH proxy value above 7 (a) and 8 (b) as a function of fragment library size. We compare standard independent embeddings of fragment selection actions (blue) with our fingerprint-based embeddings (orange) that account for the fragments' chemical structure. The number of scaffolds is reported after 2k training iterations for 3 random seeds (the solid line is the median, while the shaded area spans from minimum to maximum values). We observe that our approach greatly outperforms independent embedding when scaling to a larger action space.

## D. Docked poses of top generated molecules

**Ours (sEH, 4JNC)**

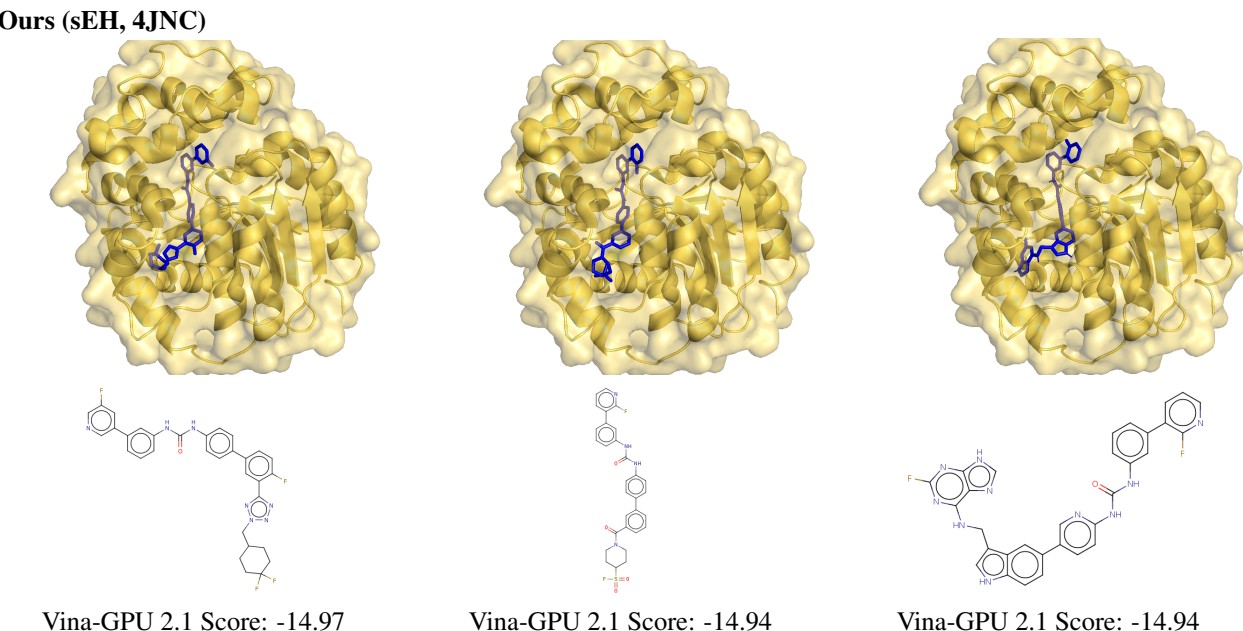

Vina-GPU 2.1 Score: -14.97    Vina-GPU 2.1 Score: -14.94    Vina-GPU 2.1 Score: -14.94

**Reference (sEH, 4JNC)**

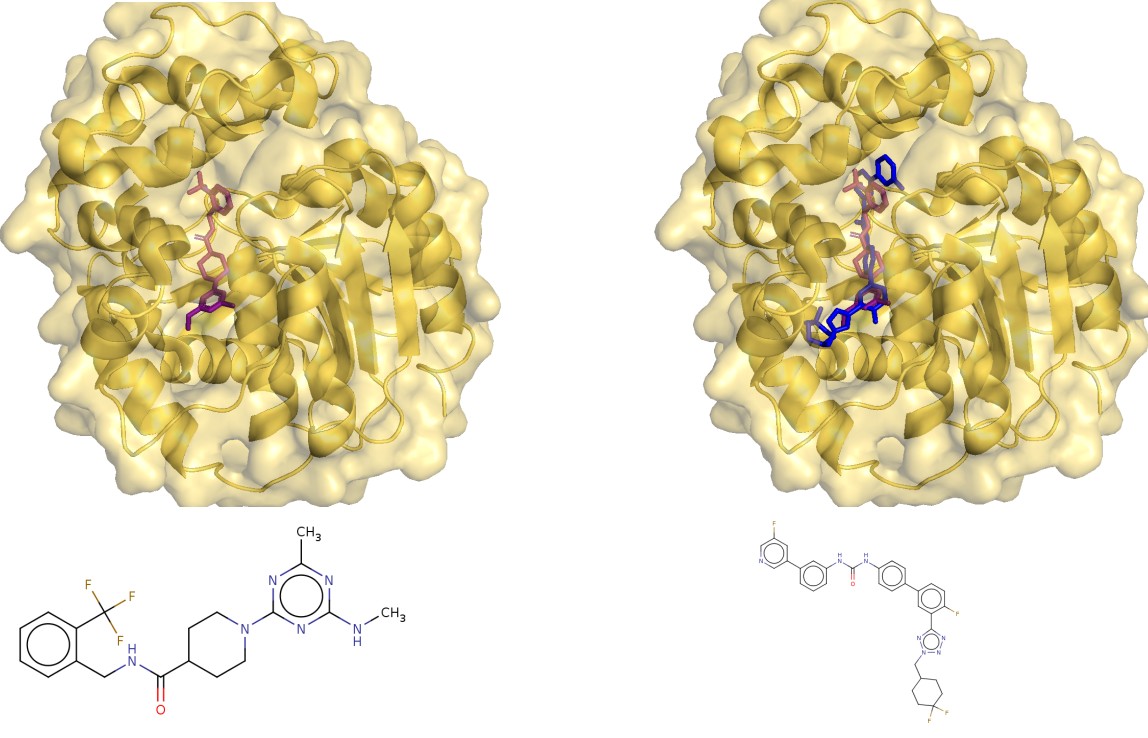

Vina-GPU 2.1 Score: -11.13    Vina-GPU 2.1 Score: -14.97

*Figure 7.* Top left to right: Top 3 generated ligand scaffolds for sEH (blue). Bottom left: Reference ligand pose (purple, PDB ID: 1LF). Bottom right: Reference ligand (purple) overlaid with top-scoring ligand (blue).

**Ours (ClpP, 7UVU)**

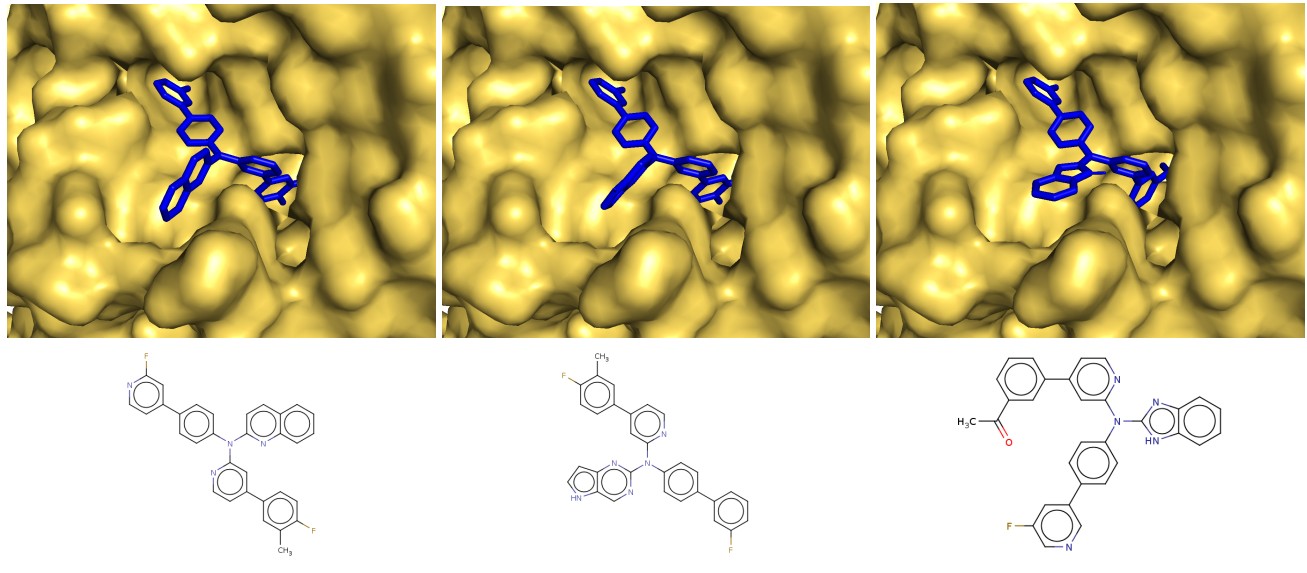

Vina-GPU 2.1 Score: -13.35          Vina-GPU 2.1 Score: -13.32          Vina-GPU 2.1 Score: -13.19

**Reference (ClpP, 7UVU)**

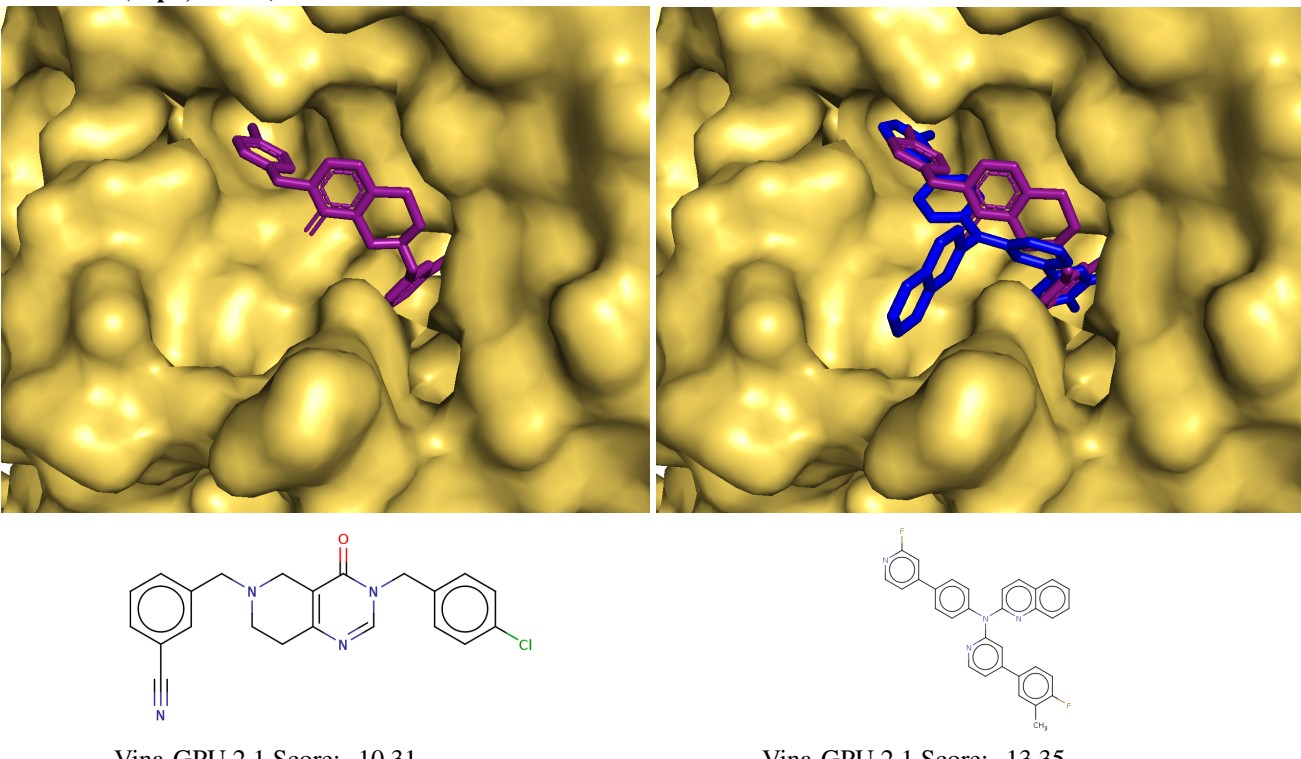

Vina-GPU 2.1 Score: -10.31                              Vina-GPU 2.1 Score: -13.35

*Figure 8.* Top left to right: Top 3 generated ligand scaffolds for ClpP (blue). Bottom left: Reference ligand pose (purple, PDB ID: OY9). Bottom right: Reference ligand (purple) overlaid with top-scoring ligand (blue).

**Ours (Mpro, 6W63)**

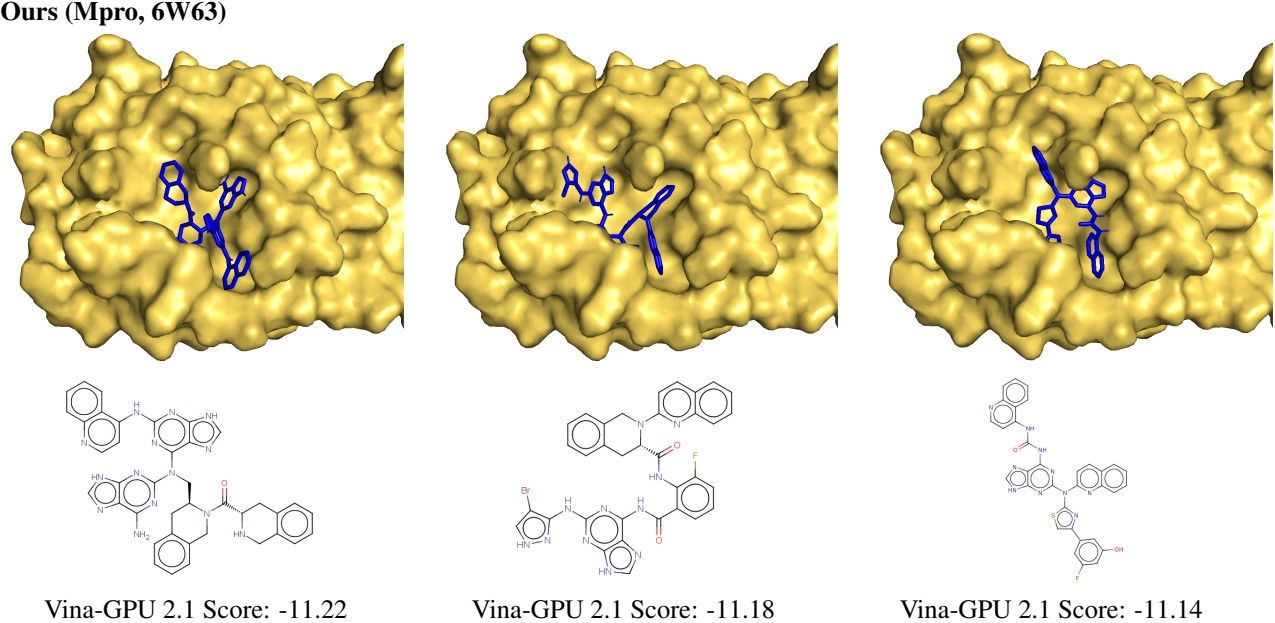

Vina-GPU 2.1 Score: -11.22          Vina-GPU 2.1 Score: -11.18          Vina-GPU 2.1 Score: -11.14

**Reference (Mpro, 6W63)**

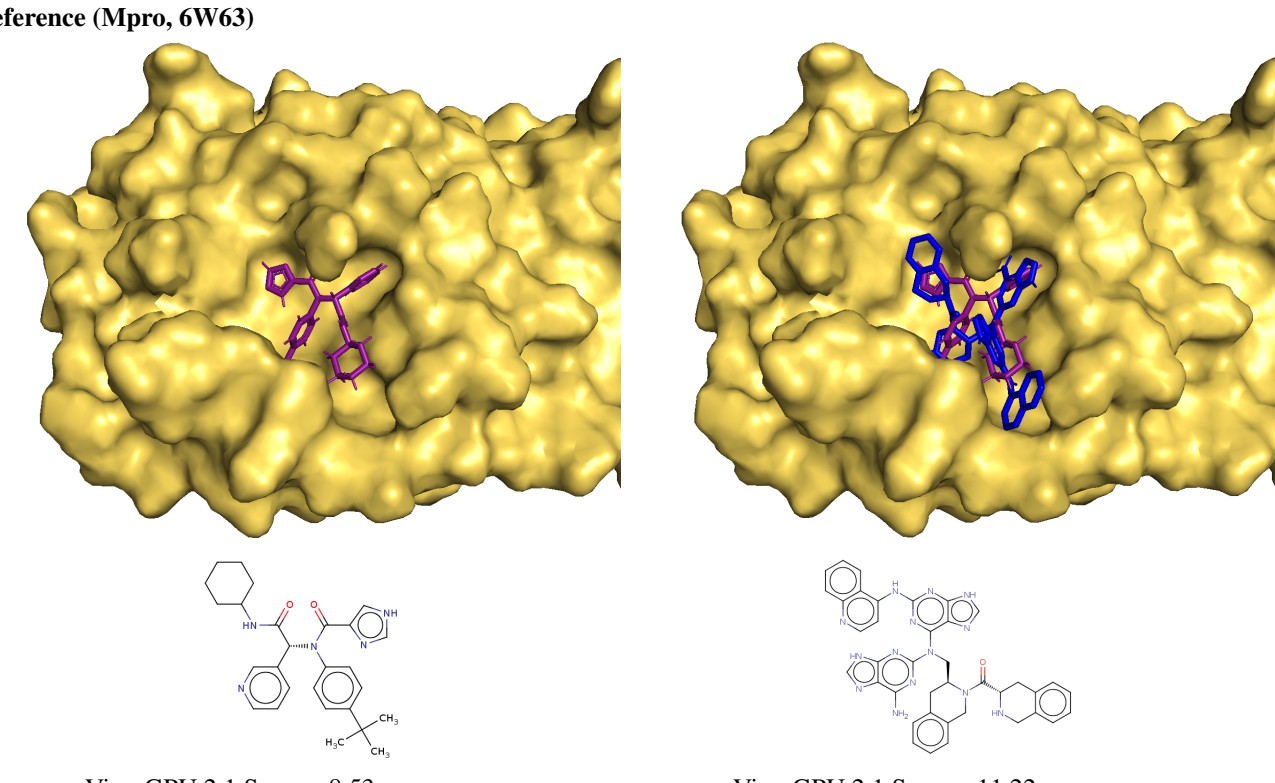

Vina-GPU 2.1 Score: -8.53                    Vina-GPU 2.1 Score: -11.22

*Figure 9.* Top left to right: Top 3 generated ligand scaffolds for Mpro (blue). Bottom left: Reference ligand pose (purple, PDB ID: X77). Bottom right: Reference ligand (purple) overlaid with top-scoring ligand (blue).

# E. Top filtered molecules for all targets

**sEH**

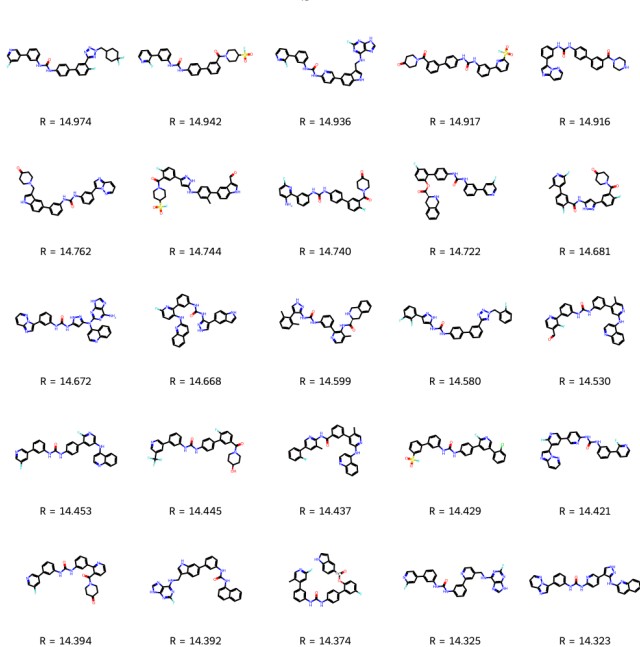

*Figure 10.* Top 25 filtered binders to sEH drawn from top 100 RGFN modes.

**ClpP**

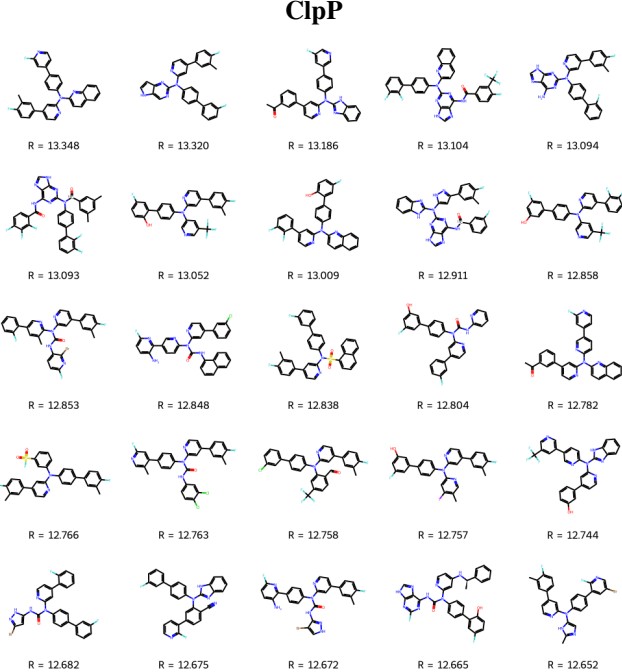

*Figure 11.* Top 25 filtered binders to ClpP drawn from top 100 RGFN modes.

**Mpro**

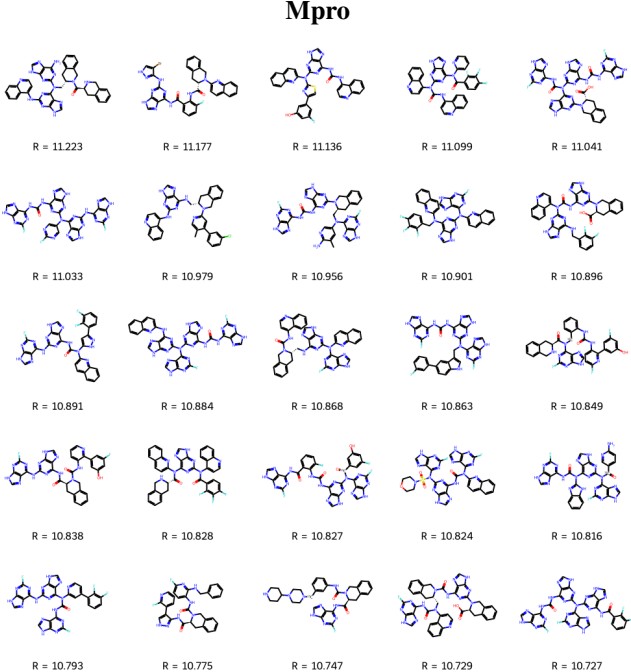

*Figure 12.* Top 25 filtered binders to Mpro drawn from top 100 RGFN modes.

**TBLR1**

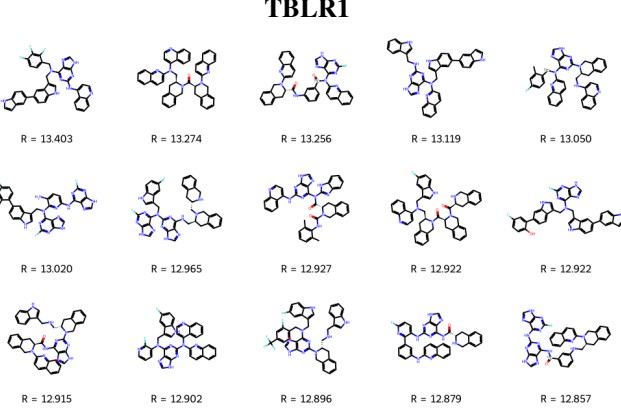

*Figure 13.* All 15 filtered binders to TBLR1 drawn from top 100 RGFN modes.