# OpenReview forum: "RGFN: Synthesizable Molecular Generation Using GFlowNets"
_ICML.cc/2024/Workshop/ML4LMS — ML4LMS Poster_

### Official Review · Reviewer_pvKp · 2024-06-12
**The paper presents an RGFN which is extension of Gflownets and strongly considers synthesizability in generative aspect of new molecules.**

**Rating:** 9
**Confidence:** 5

**Review:**

The work is clear and presents important innovation in the field of generative modeling.

---

### Official Review · Reviewer_eCsb · 2024-06-12
**Interesting idea with more applications!**

**Rating:** 6
**Confidence:** 3

**Review:**

**Summary**: The paper presents an extension of the GFlowNet framework for generating molecules by breaking the problem into a sequence of actions, specifically, the selections of chemical reactions and molecule fragments, then using a reward function on the final molecule.
The authors also discussed how the size of the search space of the generated molecules increases with more possible reactions/ fragments. I find the idea interesting and applicable to broader sorts of molecule generations.